



# GeoAI: a review of Artificial Intelligence approaches for the interpretation of complex Geomatics data

Roberto Pierdicca[1] and Marina Paolanti[2,3]

[1]Department of Civil and Building Engineering and Architecture, Università Politecnica delle Marche, Ancona, Italy
[2]Department of Information Engineering, Università Politecnica delle Marche, Ancona, Italy
[3]Department of Political Sciences, Communication and International Relations, University of Macerata, Macerata, Italy

**Correspondence:** Marina Paolanti (marina.paolanti@unimc.it)

**Abstract.** Researchers have explored the benefits and applications of modern artificial intelligence (AI) algorithms in different scenarios. For the processing of geomatics data, AI offers overwhelming opportunities. Fundamental questions include how AI can be specifically applied to or must be specifically created for geomatics data. This change is also having a significant impact on geospatial data. The integration of AI approaches in geomatics has developed into the concept of Geospatial Artificial

Intelligence (GeoAI), which is a new paradigm for geographic knowledge discovery and beyond. However, little systematic work currently exists on how researchers have applied AI for geospatial domains. Hence, this contribution outlines AI-based techniques for analysing and interpreting complex geomatics data. Our analysis has covered several gaps, for instance defining relationships between AI-based approaches and geomatics data. First, technologies and tools used for data acquisition are outlined, with a particular focus on RGB images, thermal images, 3D point clouds, trajectories, and hyperspectral/multispectral

images. Then, how AI approaches have been exploited for the interpretation of geomatic data is explained. Finally, a broad set of examples of applications are given, together with the specific method applied. Limitations point towards unexplored areas for future investigations, serving as useful guidelines for future research directions.

## 1 Introduction

Geomatics is a discipline that deals with the automated processing and management of complex 2D or 3D information. It is

defined as a multidisciplinary, systemic, and integrated approach that allows collecting, storing, integrating, modelling, and analysing spatially georeferenced data from several sources, with well-defined accuracy characteristics and continuity, in a digital format (Gomarasca, 2010).

    Nowadays, the processing of large amounts of data and information in an interdisciplinary and interoperable way relies on a growing variety of tools and data collection methods. The binomial science–technology directly connected to the geomatics

disciplines allows the continuous development of techniques for acquiring and representing data. Surveying and representation are closely linked to each other, as shown by the close connection between the disciplines traditionally associated with surveying, such as Geodesy, Topography, Photogrammetry, and Remote Sensing, and those related to representation, such as Cartography (Konecny, 2002).





Geomatics data are acquired by various systems and platforms, generating geo-spatial and spatio-temporal heterogeneous

information; indeed, the acquisition techniques provide different geomatics data, which can be images (RGB, multi- and hyper-spectral and thermal), trajectories, and point clouds. To date, existing algorithms for data processing mainly work with manual or semi-automatic approaches, since the full automation has not yet achieved greater reliability and accuracy. The resulting metric and georeferenced information are then used, catalogued, administered, displayed, and stored in a Geographic Information System (GIS) or generic databases. However, after moving into the era of big data, the analysis and practical use of

the information contained within this huge amount of data require tailored computational approaches such as Machine Learning (ML) and Deep Learning (DL) (LeCun et al., 2015). The attractive feature of AI is its ability to identify relevant patterns within complex, nonlinear data without the need for any a priori mechanistic understanding of the geomatics processes. Today, DL and AI algorithms have been successfully developed and applied in many geomatics applications (Martín-Jiménez et al., 2018; Zhang et al., 2020). According to the type of data collected, different AI methods are proposed for classification, semantic

segmentation, or object detection (Hong et al., 2020b).

## 1.1 Theoretical background, motivation and research questions

Existing reviews explore particular geomatics data approaches, generally based on ML and DL, to solve a specific issue. Examples of well-structured systematic reviews focused on RGB-D images (Guo et al., 2016; Li et al., 2018b; Zhao et al., 2019; Zhu et al., 2017), thermal images (Ali et al., 2020; Dunderdale et al., 2020; Kirimtat and Krejcar, 2018; Vicnesh et al.,

2020), point clouds (Guo et al., 2020; Li et al., 2020b; Xie et al., 2020; Zhang et al., 2019a), trajectories (Bian et al., 2018; Yang et al., 2018a; Bian et al., 2019), and hyperspectral and multispectral images (Audebert et al., 2019; Ghamisi et al., 2017; Li et al., 2019a; Signoroni et al., 2019; Yuan et al., 2021; Zang et al., 2021; Kattenborn et al., 2021) and their applications are available in the scientific literature. However, while the scientific literature recognises the importance of geomatics data processing since it covers many fields of application, there is a lack of a systematic investigation dealing with AI-based data

processing techniques. For geospatial domains, fundamental questions include how AI can be specifically applied to or must be specifically created for geospatial data. In (Janowicz et al., 2020), Janowicz et al. proposed an overview of spatially explicit AI. ML has been a core component of spatial analysis in geomatics for classification, clustering, and prediction. In addition, DL is being integrated with geospatial data to automatically extract useful information from satellite, aerial, or drone imagery (just to mention some) by means of image classification, object detection, semantic and instance segmentation, etc. The integration

of AI, ML, and DL with geomatics is broadly recognized and defined as "Geomatics Artificial Intelligence (GeoAI)".

Considering the latest achievements in data collection and processing (Grilli et al., 2017), geomatics is facing the worldwide challenge of, on one hand, reducing the need of manual intervention for huge datasets and, on the other, improving methods for facilitating their interpretation. GeoAI could represent the turning point for the entire research community, but, to the best of our knowledge, there is currently no survey on this emerging topic.

To close this gap, this review aims to provide a technical overview of the advances and opportunities offered by AI for automatically processing and analysing geomatics data. This work emphasises that, despite their specific technical require-ments, the computational methods used for these tasks can be integrated within a single workflow to optimise several steps of





interpreting complex geomatics data, regardless of the application. Considering the multidisciplinary nature of geomatics data, major efforts have been undertaken in regard to RGB-D images, infrared thermographic (IRT) images, point clouds, trajecto-

ries data (TRAJ) and multispectral imaging (MSI) and hyperspectral imaging (HSI). Initially, a literature review was conducted to understand the main data acquisition technologies and if and how AI methods and techniques could help in this field. In the following account, specific attention is given to the state of the art in AI with the selected data type mentioned above. In particular, the techniques and methods for each type of research are analysed, the main paths that most approaches follow are also summarised, and their contributions are indicated. Thereafter, the reviewed approaches are categorised and compared

from multiple perspectives, including methodologies, functions, and an analysis of the pros and cons of each category. Each technology and method reported in Figure 1 will be analysed.

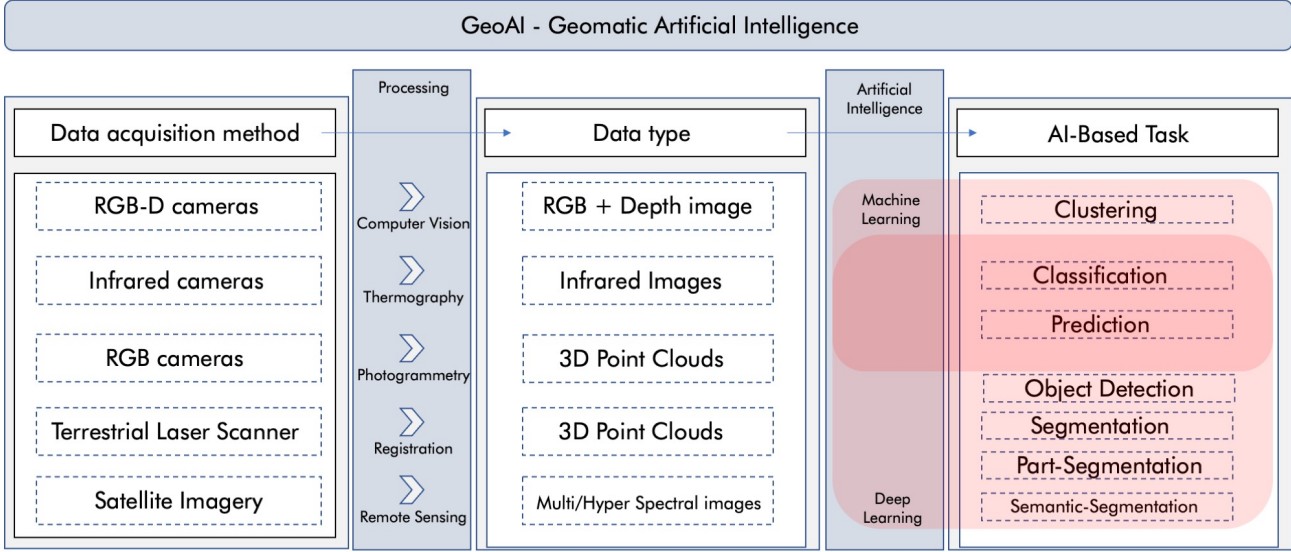

**Figure 1.** Artificial Intelligence approaches for the interpretation of complex geomatics data. Conceptualisation of the review process.

In particular, the purposes, issues, and motivations of this study were investigated to set the following research questions (RQ):

RQ1  To explore the most used methodologies in recent years for dealing with geomatics data, the following question has been

set: *Among the well established AI methods, which is the most used in geomatics?*

RQ2  To understand if the methodologies used depend on the processed data, the following question arises: *Do geomatics data influence the choice of using one methodology rather than another?*

RQ3  To provide an overview of the main tasks performed using geomatics data, the following question must be answered: *For which tasks are geomatics data used?*





RQ4  To better understand which type of geomatic data is used in different application domains, the following question arises:

*Are there relationships between application domains and geomatics data?*

## 1.2 Research strategy definition

The following sources of information were used in this study: ieeeXplore[1], Scopus[2], Sciencedirect[3], citepseerx[4], and Springer-Link[5]. A set of keywords were chosen in relation to the Remote Sensing domain and based on preliminary screening of the

research field. The keywords considered in the research initially were as follows: *geomatics data, pattern recognition, artificial intelligence, machine learning, neural networks, supervised learning, unsupervised learning, statistical methods, Active learning, Imbalanced class learning, deep learning, Convolutional Neural Networks, classification, segmentation, detection, pattern recognition, applications, remote sensing data, hyperspectral data, point clouds data, RGB-D data, thermal data, and trajectory.*

To obtain more accurate results, the keywords were aggregated. In a set of queries, the keyword *geomatics data* was combined with others related to the methodologies (ML, DL, and more), and in other sets, remote sensing data were combined with the application (classification or detection). Each query produced a large quantity of articles, which were selected based on their pertinence and year of publication. Articles considered inconsistent with the research topic and published before the year 2016 were removed from the list.

The temporal distribution of works dealing with geomatics data is shown in Figures 2 and 3. The papers considered for the review were published between the years 2016 and 2021. Figure 2 shows the temporal distribution of works dealing with AI for geomatics data. Figure 3 highlights the number of papers taken into consideration divided by the year of publication and by the type of geomatics data.

## 1.3 Paper organisation

To enhance its readability and facilitate reader comprehension, the manuscript has been structured as follows. Section 1.2 describes the methodology adopted in the choice of the articles identified and selected for the review work. Section 2 presents the related work on the application of AI methods to the geomatics data in the exam. Section 3 summarises the concepts, existing techniques and important applications of GeoAI. Section 4 describes the limitations and implications of this research, highlighting some emerging applications of AI for geomatics data analysis. Finally, Section 5 presents the implications of the

research and concluding remarks.

---

[1] https://ieeexplore.ieee.org/Xplore/home.jsp

[2] https://www.scopus.com/

[3] https://www.sciencedirect.com/

[4] https://citepseerx.ist.psu.edu/index

[5] https://link.springer.com/

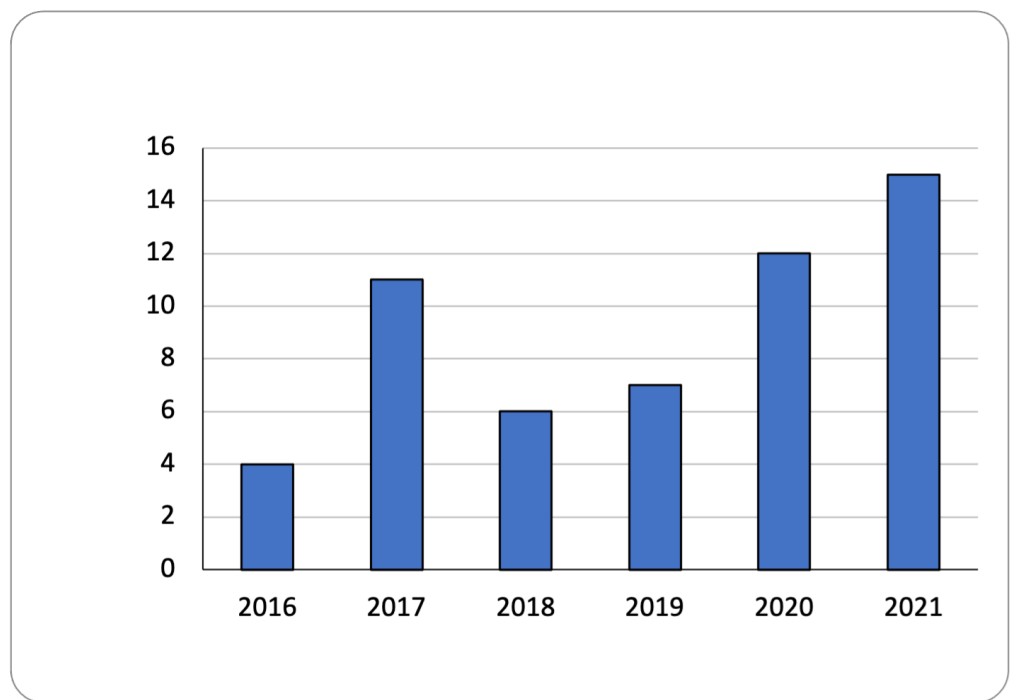

**Figure 2.** Number of publications selected per year

## 2 From traditional machine learning methods to deep learning models: Analysis of Geomatics Data

This section deepens the articles in which AI algorithms are applied for the management, processing, and interpretation of geomatics data. A set of keywords was used to perform the search phase on the channels listed in Section 1.2 and according to the taxonomy designed in Figure 1. Starting from a brief description of AI algorithms and models, a list of articles was collected
respecting the stop criterion described in the search strategy definition. The study aims at classifying research published in the field of ML and DL related to several aspects in order to compare these methods and identify their advantages and disadvantages in the application analysis.

### 2.1 Algorithms and models for GeoAI

AI aims at modelling the functioning of the human brain, and, based on the knowledge acquired, creating more advanced
algorithms. Data analysis has changed significantly with the emergence of AI and its subsets ML and DL (Paolanti and Frontoni, 2020). Over the past years, ML and feature-based tools were developed with the aim of learning relevant abstractions from data. Nonetheless, after moving into the era of multimedia big data, ML approaches have matured into deep learning approaches, which are a more efficient and powerful way of dealing with the huge amounts of data generated from modern approaches and coping with the complexities of analysing and interpreting geomatics data. DL has taken key features of the ML model and has
even taken it one step further by constantly teaching itself new abilities and adjusting existing ones (LeCun et al., 2015).



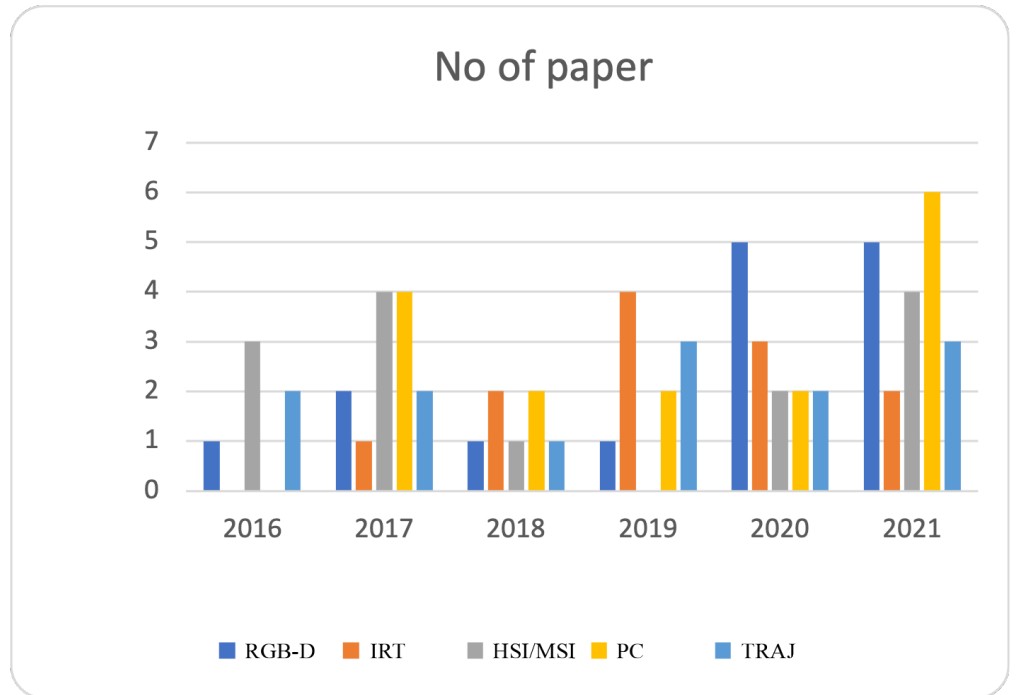

**Figure 3.** Number of selected papers for each kind of geomatic data divided by year.

AI based algorithms, especially Deep neural Networks (DNNs) are transforming the way of approaching real-world tasks done by humans. DNNs architectures are increasingly being adopted in geomatics due to their competence to learn relevant abstractions from data. At first, these models were considered as "black box" operators, but as their popularity has grown they need to be interpretable and explainable xiao2018deep, (Elhamdadi et al., 2021), (Fuhrman et al., 2021).

The main geomatics tasks solved with ML and DL models can be summarised as follows:

– Clustering (Shi and Pun-Cheng, 2019);

– Classification and Prediction (Jiang, 2018);

– Object Detection (Li et al., 2020a);

– Segmentation (Minaee et al., 2021);

– Part Segmentation (Adegun et al., 2018);

– Semantic Segmentation (Yuan et al., 2021).

Clustering is a process of grouping homogeneous elements, based on some characteristics, in a dataset. This operation in everyday life has an unlimited number of applications and is put into practice every time any grouping is carried out (Boongoen and Iam-On, 2018).





The various clustering methods include the following:

- *Connection method*, such as Linkage, which is a hierarchical method suitable for grouping both variables and observations (single linkage, based on the minimum distance; complete linkage, based on maximum distance; and average linkage, based on average distance).

- *k-means method*, which is a non-hierarchical and vector quantisation method that partitions n observations into k clusters,
in which each observation belongs to the cluster with the nearest mean (cluster centres), working as a prototype of the cluster.

- *spectral cluster*, which is an approach with origins in graph theory, where the method is used to classify communities of nodes in a graph based on the edges connecting them. The process is adaptable and allows clustering non-graph data.

Classification is the process of learning a certain target function f, which maps an input vector $x$ to one of the predefined
labels $y$. The target function is also referred to as the classification model (Tan et al., 2016).

A classification model generated through a learning algorithm must be able to adapt correctly to the input data but also, and more importantly, to correctly predict record class labels that it has never seen before. That is, the key objective of the learning algorithm is to build models with good generalisation skills.

Object detection is an important problem that consists of identifying instances of objects within an image and classifying
them as belonging to a certain class (e.g. humans, animals, or cars) (Li et al., 2020a). The goal is to develop computational techniques and models that provide one of the basic elements necessary for computer vision applications, specifically knowing which objects are in an image. Object detection is the basis of many applications for computer vision, such as instance segmentation, image captioning, and object tracking. From an application point of view, it is possible to group object detection into two categories: "general object detection" and "detection applications" (Liu et al., 2020). For the first, the goal is to investigate
methods for identifying different types of objects using a single framework in order to simulate human vision and cognition. In the second case, we refer to the recognition of objects of a certain class under specific application scenarios: this is the case of applications for pedestrian detection, face detection, or text detection. Currently, the models for object detection can be divided into two macro categories: two-stage and one-stage detectors. Two-stage models divide the task of identifying objects into several phases, following a "coarse-to-fine" policy. One-stage models complete the recognition process in a single step
with the use of a single network.

The problem of image segmentation is a topical research field due to its numerous applications in different fields, from signal processing at the industrial level to the biomedical sector, where it can represent a valid technique for facilitating the reading and quantitative evaluation of the outputs coming from complex diagnostic tools (e.g. magnetic resonance imaging) (Fu and Mui, 1981). Segmentation is a process of dividing an image into separate portions (segments) that are groupings of neighbouring
pixels that have similar characteristics, such as brightness, colour, and texture. The purpose of segmentation is to automatically extract all the objects of interest contained in an image; it is a complex problem due to the difficult management of the multitude of semantic contents (Sultana et al., 2020).



According to (Naha et al., 2020), there are several recent papers that have demonstrated that the use of DL approaches yields very good performance on object part segmentation considering both rigid and non-rigid objects.

As mentioned earlier, an image segmentation model enables partitioning an image into different regions representing the different objects. We talk about semantic segmentation when the model is also able to establish the class for each of the identified regions. In other words, carrying out a semantic segmentation means dividing an image into different sets of pixels that must be appropriately labelled and classified in a specific class of belonging (e.g. animals, humans, buildings).

Semantic segmentation can be a useful alternative to object detection, as it allows the object of interest to cover multiple
areas of the image at the pixel level. This technique detects irregularly shaped objects, unlike object detection, where objects must fit into a bounding box (Felicetti et al., 2020).

Semantic segmentation of point clouds is also an important step for understanding 3D scenes. For this reason, it has received increasing attention in recent years and a lot of AI approaches have been proposed in order to automatically identify objects (Zhang et al., 2019a; Malinverni et al., 2019; Paolanti et al., 2019).

**2.2    Geomatics: a fundamental source of data**

This Section aims to classify the various types of sensors for data acquisition and describe their characteristics. The classification scheme was selected according to the acquisition device and data features, considering the following: i) the output data structuring, ii) the active/passive sensors, and iii) the type of actuation. The main distinction in this review is the type of sensor (i.e. if the acquisition system is supplied with a laser sensor or on a vision sensor, such as a camera). It is fair to state that this
is not an exhaustive list of all possible geomatic techniques; rather, it attempts to embrace all the sensors that generate data for which interpretation, given their complexity, requires the aid of statistical learning-based approaches.

A revolutionary turning point in terms of the concept of geomatics was brought by the research paper titled "Geomatics and the new Cyber-Infrastructure" (Blais and Esche, 2008). In that paper, the authors state that geomatics deals with multi-resolution geospatial and spatio-temporal information for all kinds of scientific, engineering, and administrative applications.
This sentence can be summarised as follows: geomatics is far more than the concept of simply measuring distances and angles. Multi-resolution geospatial data (and metadata) refer to the observations and/or measurements at multiple scalar, spectral, and temporal resolutions, such as digital imagery at various pixel sizes and spectral bands that can provide different seasonal coverage. Surveying still plays a leading technological role, but it has evolved in new forms: positioning and navigation can be obtained with several devices that were not properly conceived to accomplish these tasks; topographical mapping, once
conducted with bulky instruments requiring complex computations on the part of workers, has now become a byproduct of Geospatial or GIS; digital images, obtained with different sensors (from satellite images to smartphones), can be used to accomplish both the tasks of classifying the environment and making virtual reconstructions. Survey networks and photogrammetric adjustment computations have largely been replaced by more sophisticated digital processing with adaptive designs and implementations or ready-to-use equipment, such as Terrestrial Laser Scanners (TLS).
Analysis tasks can be performed at a regional level thanks to the use of high-resolution images from satellite or aerial images; inferring information is possible through land usage classification, and the shape can be described using ranging techniques





like LiDAR and radar pulse. The possibilities offered by new acquisition devices for dealing with architectural-scale complex objects are numerous. Low-cost equipment (cameras, small drones, depth sensors, and so on) are capable of accomplishing reconstructions tasks. Of course, there is accuracy must also be considered. In fact, geo-referencing complex models requires
more sophisticated and accurate data sources like a GNSS (Global Navigation Satellite System) receiver or TLS. In the case of small objects or artefacts, terrestrial imagery and close-range data are the best solutions for obtaining detailed information about them. In the following, we report the main areas of application that are closely related to geomatics, which emerged from the previous analysis: natural environment; quality of life in rural and urban environments; predicting, protecting against, and recovering from natural and human disasters; and archaeological site documentation and preservation. In sum, geomatics is
able to cover the spectrum of almost every scale (Böhler and Heinz, 1999); while though there is no panacea, the integration of all these data and techniques is definitely the best solution for 3D surveying, positioning, and feature extraction.

### 2.2.1 RGB-D Cameras

Before Microsoft Kinect was launched in November 2010, collecting images with a depth channel was a burdensome and expensive task. Using depth as an additional channel alongside the RGB input has the scale variance problem present in image
convolution-based approaches. In the last few years, there have been attempts to combine the increasing popularity of depth sensors and the success of learning approaches, such as ML and then DL (Chu et al., 2018; Wang et al., 2021). RGB-D cameras generate a color representation (Red, Green, and Blue) of a scene and allow reconstruction of a depth map of the scene itself (Han et al., 2013; Liciotti et al., 2017). The depth map is an image M of $MxN$ dimension, in which each pixel p(x,y) represents the distance in the 3D scene of the point (x,y) from the sensor that generated it (Fu et al., 2020; Jamiruddin
et al., 2018). The use of depth images compared to RGB or BW (Black and White) images provides information about the third dimension and simplifies many computer vision and interaction problems, such as: i) background removal and scene segmentation; ii) tracking of objects and people; iii) 3D reconstruction of the environment; iv) recognition of body poses; and v) implementation of gesture-based interfaces (Han et al., 2013). To determine the depth map, the considered devices use a pattern projection technique. This involves a stereo vision system consisting of a projector and camera pair in order to define
an active triangulation process. ii) projection of periodic 2D patterns and study of their deviation when they reach objects; and iii) projection of pseudo-random 2D patterns.

For a semantic segmentation task involving urban/rural scenes, the work of (Li et al., 2017a) proposes a method based on RGB-D images of traffic scenes and DL. They use a new deep fully convolutional neural network architecture based on modifying the AlexNet (Krizhevsky et al., 2012) network for semantic pixel-wise segmentation. The RGB-D dataset is built
by the cityscapes dataset (Cordts et al., 2016), which comprises a large and diverse set of stereo video sequences of outdoor traffic scenes from 50 different cities. The original AlexNet is modified since they perform a batch normalisation operation on the output of each convolutional layer, and during the experimental phase they find that this modification improves the segmentation accuracy. The modified version of AlexNet is used as the encoder network of the architecture. During the test, they evaluate the semantic segmentation performance of the proposed architecture, comparing the results obtained with RGB-D



230 images as input and only RGB images as input. The experimental results show that the use of the disparity map increases the semantic segmentation accuracy, achieving good real-time performance and segmentation accuracy.

To semantically segment RGB-D frames collected in commercial buildings and to recognise all component classes of buildings, a DL artificial neural network method is used in (Czerniawski and Leite, 2020). The purpose is to demonstrate that the proposed method is able to semantically segment RGB-D images into 13 classes of components even if the training dataset is

235 very small. The dataset was purposely built and manually annotated using a common building taxonomy in order to provide complete semantic coverage. The supervised neural network that is used is DeepLab (Chen et al., 2017), a state-of-the-art model for semantic segmentation of images that assigns a semantic label to each pixel of the image. To demonstrate the validity of the approach, the authors compare the performance with several state-of-the-art DL methods used for building object recognition.

240 Finally, RGB-D images have been exploited to fulfil localization tasks (Zhang et al., 2021) and 3D object part segmentation (Zhuang et al., 2021)

### 2.2.2 Infrared Cameras

Thermography, or thermovision, is a non-invasive, simple, and precise investigation system that provides real-time infrared images of any object opaque to this radiation, allowing the visualisation (and quantitative representation) of its surface temper-

245 ature (Gade and Moeslund, 2014). The images are usually represented in false color scales, in which a certain color corresponds to a certain temperature and is not the real color of the object.

Infra Red Termography (IRT) is a well-known method of examination, which is useful because if is safe, painless, non-invasive, easy to reproduce, and has low running costs. IRT combined with AI-based automated image processing can easily detect and analyse damage or other failures in images (Kandeal et al., 2021). Despite the literature proposes approaches based

250 on the single RGB data Espinosa et al. (2020), IRT images proved to be more reliable.

The classification of defects in thermal images through an initial prevention mechanism is dealt with in the work of (Ullah et al., 2017), which uses an artificial neural network architecture, specifically multi-layered perceptron (MLP), for this task. The system classifies the thermal conditions of components into two classes: "defect" and "non-defect". They initially extract statistical first- and second-order features departing from thermal sample images. To increase the classification performance,

255 they augment MLP with the graph cut, obtaining better performance in the identification of defects and the classification of the images.

The same application is considered in the paper of (Nasiri et al., 2019), in which the authors propose a convolutional neural network architecture to automatically detect faults and monitor equipment operations of a cooling radiator. They consider infrared thermal images and a DL architecture that has the task of feature extraction and classification of six conditions of the

260 radiator. The architecture is constructed based on a VGG-16 structure, followed by batch normalisation, dropout, and dense layers. During the experimental phase, they compare the classification performance with other traditional artificial neural networks, demonstrating high performance and accuracy in various working conditions. In the work of (Ullah et al., 2020), a novel model is proposed that detects an increase of temperature in high-voltage electrical instruments to promptly intervene





to avoid equipment failure that could damage the system. It is important that any anomalies are detected and eliminated. In
this context, the authors identify faults and anomalies in IRT images using a combined DL architecture. The infrared thermal
images are the input of a convolutional neural network for the feature extraction task. Then, the features vector is the input of
five different ML models (RF, SVM, J48, NB, BayesNet), which are selected to categorise the performance in the classification
task into defective and non-defective classes. The experimental results demonstrate that the best classifier is the RF classifier,
which is the best for discriminating the binary classification.

Classification of faults in electrical equipment is considered in the work of Duan et al. (Duan et al., 2019). They use an
artificial neural network to automatically classify defects as water, oil, and air, which can reduce the performance of some
materials. Through a quantitative comparison they demonstrate that the approach that uses coefficients as features provides
better performance than the one using raw data.

Finally, another interesting method is proposed by (Chellamuthu and Sekaran, 2019), which uses a deep neural network
to classify parts of infrared images into two classes: defect and non-defect. They intend to evaluate and monitor the parts of
electrical equipment to identify thermal defects at an early stage in order to promptly intervene to avoid worse damage. First,
the segmented thermal images are considered. Then, based on the optimal features, the feature extraction procedure follows.
The optimal feature extraction is obtained using the Opposition-based Dragonfly Algorithm (ODA). The experimental results
demonstrate that the proposed approach obtains better accuracy performance than other classification methods.

Defect detection in infrared images of photovoltaic (PV) modules is addressed in the works of (Akram et al., 2020) (Pierdicca
et al., 2018) and  (Luo et al., 2019). The increase in the amount of PV installations makes automatic monitoring methods
important since manual/visual inspection has several limitations. In this context, these works propose a method based on a
DL algorithm that is able to automatically identify defects in infrared images on PV modules. The main approahes used are
visual geometry group-Unet (VGG-Unet) and mask region-based convolutional neural network (Mask R-CNN) architecture
that simultaneously performs object detection and instance segmentation (Pierdicca et al., 2020a)

PV module faults are also classified in the work of (Li et al., 2018a), with the aim to propose a new method for automatically
classifying defects in infrared thermal images.

Defect detection is the focus of the work of (Gong et al., 2018), in which the authors aim to identify anomalies in electrical
equipment implementing a model based on DL. The implemented defect identification models are InceptionV2 and Inception
Resnet V2. The performance of the method is also evaluated for infrared images with artificial defects.

Finally, IRT images are also used to detect faults in infrared thermal images of composite materials used in aircraft, vehicles,
and several industries exploiting their mechanical properties (Bang et al., 2020), and building monitoring (Al-Habaibeh et al.,
2021).

### 2.2.3  Digital Photogrammetry and Terrestrial Laser Scanning

Photogrammetry is a technique that enables metrically determining the shape, size, and position of an object having two distinct
photographic frames that should be central projections of the object itself (Baqersad et al., 2017). As well, 3D laser scanning
technology (Lemmens, 2011) has been widely used in the engineering and construction industries. 3D laser scanners work on



the principles of Lidar (Light Detecting and Ranging) by emitting a laser pulse, which hits a target and subsequently returns to the sensor (Liscio et al., 2018) (Di Stefano et al., 2021).

The points captured are called a point cloud, which is then exported into laser scanning software that can create fully coloured 3D models that allow for point-to-point measurements and excellent visualisation of the scene.

The use of ML and DL techniques for point cloud classification and semantic segmentation was successfully investigated in the last decade in the geo-spatial environment (Weinmann et al., 2015; Qi et al., 2017a; Özdemir and Remondino, 2019). Several methods have been recently proposed (Shen et al., 2021; Xiao et al., 2021; Geng et al., 2021), and in the following a

detailed review of the main approaches in the Geomatics filed is reported.

The pioneer DL algorithm that processes 3D point clouds is (Qi et al., 2017a). It automatically classifies and performs the semantic segmentation directly on the point clouds. They consider an architecture that first analyses the features of the single points and then identifies them globally. However, this architecture does not capture local geometries, so an optimisation of this methodology is presented in (Qi et al., 2017b). In this paper, to learn local features by exploiting the metric space distances, a

hierarchical grouping is considered. For local neighbourhoods, the experimental phase shows improvement results compared with other state-of-the-art architectures.

To handle 3D point clouds with spectral information acquired by Lidar systems, the work presented by (Yousefhussien et al., 2018) uses a method based on DL algorithms. They propose a modified version of PointNet (Qi et al., 2017a) to obtain a model able to operate with complex 3D data acquired from overhead remote sensing platforms using a multi-scale approach.

Their DL network can directly deal with unordered and unstructured point clouds without modifying the representation and losing information. Moreover, to demonstrate the accuracy of their method, they present a performance comparison with other state-of-the-art methods. Papers like (Zhang and Zhang, 2017; Wang and Ji, 2021; Lee et al., 2021) make extensive use of approaches based on DL for semantic parsing of 3D point clouds of urban building scenes.

In (Zhang et al., 2018), the problem of semantic segmentation of 3D scenes on a large scale is tackled by considering a

fusion between 2D images and 3D point clouds. The authors create a Deeplab-Vgg16 high-resolution model (DVLSHR) based on Deeplab-Vgg16 and the Deep Visual Geometry Group (VGG16), which is successfully optimised by training seven deep convolutional neural networks on four reference datasets. The preliminary segmentation is made using 2D images, which are then mapped into 3D point clouds, taking into account the relation among the images and the point clouds. Subsequently, based on the mapping, the physical planes of buildings are extracted from the 3D point clouds.

In the field of Digital Cultural Heritage (DCH), the work of (Pierdicca et al., 2020b) uses an improved version of DGCNN (Wang et al., 2019) that adds meaningful features, such as normal and colour. The aim is to semantically segment 3D point clouds to automatically interpret the architectural parts of buildings and obtain a useful framework for documenting monuments and sites. They use a novel dataset comprising both indoor and outdoor scenes, which are manually labelled by experts and which belong to different historical periods and styles (Matrone et al., 2020b). Extensive experiments on the purposely created dataset

show the efficiency of the optimised architecture, and the results are compared with those of other state-of-the-art models. The authors have also extended the proposed approach by comparing the DL approach with a ML based one and by the improvement of DGCNN with other relevant features (Matrone et al., 2020a).



A DL-based framework for automatically extracting, classifying, and completing road markings from three-dimensional mobile laser scanning (MLS) point clouds is presented by (Wen et al., 2019). A modified version of the UNet architecture is

used to extract road markings. For classification, a method based on clustering and convolutional neural networks is developed, and it is more efficient with different sizes. Finally, to complete the road marking, a method based on a conditional generative adversarial network (cGAN) is used, which is more effective since it considers the continuity and regularity of the lane lines. The dataset consists of three scenes, highways, urban roads, and underground parking, with raw point clouds and labelled road marking ground truths.

In the context of urban/rural scenes, the paper of (Yang et al., 2017) proposes a method for semantically labelling 3D point clouds acquired by an airborne laser scanner using an approach based on DL. A point-based feature image generation method extracts local geometric features, global geometric features, and full-waveform features from 3D point clouds, transforming them into an image. Then, the feature images are the input of a convolutional neural network for semantic labelling. Finally, to compare the performance of the proposed approach with state-of-the-art methods, they test the framework, using other publicly

available datasets, achieving a high level of overall accuracy with the proposed network.

To solve a similar issue, the paper of (Wang et al., 2019) and (Can et al., 2021) uses a novel convolutional neural network called Dynamic Graph CNN (DGCNN), which includes a new module called EdgeConv that acts on graphs dynamically computed in each layer of the network. The EdgeConv module incorporates local neighborhood information, can be applied to learn global shape properties, and captures semantic characteristics in the original embedding. To demonstrate the performance

of the proposed model, the authors use different public datasets: ModelNet40, ShapeNetPart, and S3DIS. Moreover, they compare the results with other models based on DL, obtaining better results in terms of accuracy.

To minimise the large number of point clouds needed to classify urban objects, a solution is proposed by (Balado et al., 2020). The problem that they intend to address is (Balado et al., 2020). They use convolutional neural networks to convert point clouds into pc images, taking into account that acquiring and labelling point clouds is more expensive and time consuming than the

corresponding image. They generate several sample images per object (point clouds) by means of multi-view and combine pc images with images derived from online datasets: ImageNet and Google Images. The DL algorithm chosen is InceptionV3. To validate the proposed methodology, they also consider the IQmulus & TerraMobilita Contest dataset, obtaining correct classification with few samples.

Complex forest scenes represented by 3D point clouds are classified using a method based on DL in the work of (Zou et al.,

2017). A new voxel-based DL method classifies species of trees, using 3D point clouds of forests as input and consisting of three phases: individual tree extraction, feature extraction, and classification using DL. Moreover, two different datasets acquired using terrestrial laser scanning systems are used. Then, to evaluate the performance and demonstrate the effectiveness of the proposed method, they also compare it with other classification methods for 3D tree species. Other interesting works worth to metion in this field are (Chen et al., 2021; Pang et al., 2021).





### 2.2.4 Remote sensing: multi and hyper spectral data

Remote sensing (Toth and Jóźków, 2016) is a technical-scientific discipline that allows obtaining quantitative and qualitative information and measuring the emission, transmission, and reflection of electromagnetic radiation from surfaces and bodies placed at a very high distance from an observer. Recently, ML approaches as part of the AI domain and its DL subset have become increasingly important in MSI and HSI remote sensing analysis (Yuan et al., 2021; Zang et al., 2021). Several works have been proposed with the aim of expediting time-consuming processes (Zhu et al., 2017).

In the following, different papers are presented to solve the classification task of HSI/MSI images of urban/rural scenes, mainly using DL algorithms.

The only paper considered that uses an approach based on ML is (Sharma et al., 2017). The aim is to evaluate the performance of different supervised ML classifier in discrimination of six vegetation physiognomic classes. They use supervised approaches with different model parameters and demonstrate that Random Forests classifier provides the greatest accuracy and kappa coefficient.

The work of (Zhong et al., 2017) proposes a system that classifies hyperspectral images using a supervised model based on DL. The input of the spectral-spatial residual network (SSRN) is represented by 3D raw cubes. Through identity mapping, each of the 3D convolutional layers is connected by the residual blocks. Then, to improve the classification accuracy and the learning process, a batch normalisation algorithm is used on each convolutional layer. The dataset is made up of agricultural, rural–urban, and urban hyperspectral images. The qualitative and quantitative experimental results indicate that the proposed framework achieves good classification accuracy. Many other papers adopt similar approaches, like Mendili et al. (2020) for LC/LU classification, Audebert et al. (2018) for semantic labelling, shadow detection Movia et al. (2016), precision farming Zheng et al. (2020).

To deal with the hyperspectral image classification problem, Yang et al. (Yang et al., 2018b) present a method for increasing the classification performance, exploiting both the spatial context and spectral correlation, although in general only the spatial context is considered. Specifically, they consider and evaluate the performance of four convolutional neural networks: 2DCNN, 3DCNN, recurrent 2DCNN, and recurrent 3DCNN. Six open-access datasets are used for classification. Moreover, to demonstrate that DL methods provide better performance in the classification task, four architectures are compared with other traditional methods.

In addition, Wu et al. (Wu and Prasad, 2017) propose a method for classifying hyperspectral images using DL methods. They highlight the need to have a large amount of labelled data for training, and to solve this problem they propose a semi-supervised DL approach that requires limited labelled data and a large amount of unlabelled data, which they use with their pseudo labels (cluster labels) to pre-train a deep convolutional recurrent neural network, which they fine-tune using a smaller amount of labelled data. Moreover, to use spatial information they implement a constrained Dirichlet process mixture model (C-DPMM) for semi-supervised clustering, also deriving a variational inference model.

The paper of (Zhao and Du, 2016) proposes a novel classification framework based on a spectral–spatial feature (SSFC) that uses dimension reduction and DL methods to extract spectral and spatial features, respectively. Spectral feature extraction is





applied to high-dimensional hyperspectral images using a local discriminant algorithm, while a convolutional neural network is implemented to determine high-level spatial features. Finally, the multiple features extracted considering jointly spectral and spatial features are used to train the multiple-feature-based classifier for image classification. To demonstrate the performance of the SSFC classifier, they compare the results with those of other traditional classification methods.

A target detection for hyperspectral images using a deep convolutional neural network is proposed in (Li et al., 2017b). To train this multi-layer network, a high amount of labelled samples is needed, but for target detection, few labelled targets are available. Hence, to enlarge the dataset, they further generate pixel pairs. In the experimental phase, two cases are considered: in the first, for anomaly detection, using similarity measurements, a convolutional neural network classifies different pixel-pairs, obtained by combining the centre pixel and its surrounding pixels; in the second, for supervised target detection, a convolutional neural network classifies different pixel pairs obtained by combining the testing pixel and the known spectral signatures.

The aim of (Liu et al., 2016) is the classification of hyperspectral images using active DL. As obtaining well-labelled samples for remote sensing applications is very expensive, they consider weighted incremental dictionary learning. The algorithm selects samples by maximising two selection criteria: representativeness and uncertainty. Moreover, the network is actively trained to select training samples in each iteration. To validate the proposed architecture, during the experimental phase they compare the performance with other classification algorithms that use active learning.

In (Chen et al., 2016), the argument concerns the classification task of hyperspectral data. The authors propose a DL approach to elaborate hyperspectral images. In particular, they combine a novel feature extraction (FE) and image classification architecture based on a deep belief network (DBN) to obtain high classification accuracy. During the experimental phase, they demonstrate that the framework provides encouraging classification results compared with other state-of-art methods. Moreover, they demonstrate the great potential of DL methods for classifying hyperspectral images, even confirmed in more recent works (Xu et al., 2021).

The paper proposed by (Hong et al., 2020b) aims to demonstrate that the use of a framework based on DL, in particular a cross modal DL framework called X-ModalNet, provides good results for classification tasks of multispectral imagery (MSI) and synthetic aperture radar (SAR) data. The architecture consists of three well-designed modules: a self-adversarial module, interactive learning module, and label propagation module. During the experimental phase, the authors compare the classification performance with other state-of-the-art methods, demonstrating significant improvement.

In the paper of (Hong et al., 2020a), a framework based on DL is presented to classify hyperspectral data. In particular, convolutional neural networks and graph convolutional networks are used to classify hyperspectral images. The authors develop a new minibatch graph convolutional network to solve the problem of huge computational costs in large-scale remote sensing problems. The mini graph convolutional network infers out-of-sample data without the need to retrain the networks and improves the classification performance. Since convolutional and graph convolutional networks extract different types of features, they are fused based on three strategies (additive fusion, elementwise multiplicative fusion, and concatenation fusion) to increase the classification performance. The experimental results from three different datasets demonstrate that the





use of mini graph convolutional networks provides better performance than graph convolutional networks as well as combined convolutional and graph convolutional GCN models.

Worthmentioning is the work presented by Li et al. (2019b) detects changing in synthetic aperture radar (SAR) images. The authors use a DL architecture, specifically a convolutional neural network trained to obtain a classifier able to distinguish modified pixels from unmodified pixels. This task is very important when disasters occur where it is difficult to obtain prior knowledge. To address this issue, they modify a supervised training process into an unsupervised learning process. Moreover, this method does not require image preprocessing and a filtering operation for SAR images. A convolutinal neural network

makes use of the spatial feature and neighbourhood information of pixels to learn the hierarchical features of the images and implement an end-to-end framework.

### 2.2.5 GNSS positioning

GNSS (Global Navigation Satellite System) is the positioning system based on the reception of radio signals transmitted by various constellations of artificial satellites (Groves, 2015). Modern GPS receivers have achieved very low costs. The market

now offers low-cost solutions for all uses, which are effective not only for satellite navigation but also for civil uses, for monitoring mobile services, and for territorial control. Consequently, trajectory forecasting has been a field of active research owing to its numerous real-world applications, thanks to the ever increasing availability of GNSS data, for both pedestrians (Kothari et al., 2021) and veichles (Siddique and Afanasyev, 2021).

     The aim of the paper of (Endo et al., 2016) is to addresses the problem of extracting the characteristics that estimate users'

transport modes based on their movement trajectories. To compensate for a lack of handcrafted functionality, they propose a method that automatically extracts additional functionality using a deep neural network. A classification model is constructed in a supervised manner using both deep and handcrafted characteristics. The effectiveness of the proposed method is demonstrated through several experiments using two real datasets, comparing the accuracy with that of previous methods.

     Another paper (Habtemichael and Cetin, 2016) presents a non-parametric, data-driven methodology for short-term traffic

prediction based on recognising similar traffic patterns, employing an advanced K-closer algorithm. Additionally, winsorisation of neighbours is implemented to reduce the consequences of predominant candidates, and the rank exponent is applied to aggregate candidate values. The robustness of the proposed method is demonstrated by implementing it on large datasets derived from different regions and comparing the performance with advanced time series models, such as the SARIMA and Kalman Filter adaptive models proposed by others. Furthermore, the effectiveness of the proposed advanced K-Nearest Neigh-

bour (k-NN) algorithm is evaluated for multiple prediction stages, and its performance is also tested with data with missing values. This study provides strong evidence showing the promise of a non-parametric, data-driven method for short-term traffic prediction.

     Obtaining knowledge from the GPS tracks of human actions is the topic of the work of (Jiang et al., 2017). The author presents TrajectoryNet, a neural network architecture for point-based trajectory classification to infer real-world human trans-

port modes from GPS tracks. A new representation is developed that includes the original feature space into another space, which can be recognised as a form of base expansion, to overcome the challenge of capturing the underlying latent factors





in the low-dimensional and heterogeneous feature space imposed by GPS data. A classification accuracy greater than 98% is achieved for identifying four types of transport modes, exceeding the performance of existing models, without further sensory data or prior knowledge of the location.

According to (Xiao et al., 2017), transport mode identification can be used in a variety of applications, including human behaviour research, transport management, and traffic control (Yang et al., 2021). In this paper, a learning set-based method is presented to infer hybrid modes of transport employing only GPS data. First, in order to distinguish between diverse modes of transport, a statistical approach is used to produce global features and extract different local features from sub-trajectories after trajectory segmentation before these features are combined in the classification step. Second, to obtain better performance,

tree-based ensemble models (Random Forest, Gradient Boosting Decision Tree, and XGBoost) are used instead of traditional methods (K-Nearest Neighbor, Decision Tree, and Support Vector Machines) to classify the different transport mode tools.

Correct detection in public transport modes is a fundamental task in smart transport systems according to (James, 2020). Hence, the aim is to utilise GPS trajectories of random lengths to produce efficient travel mode results in global and online classification scenarios. Raw GPS data are processed to calculate preliminary movement and displacement properties, which

are fed into a tailored deep neural network. The results show that the approach can significantly exceed state-of-art travel mode identifications with the same dataset with little computation time. Moreover, an architecture test is performed to determine the best-performing structure for the proposed mechanism.

According to the work of (Dabiri et al., 2019), recognising passenger transport modes is important for many issues in the transport field, such as travel demand analysis, transport planning, and traffic management. The paper aims to classify travellers'

modes of transport based only on their GPS trajectories. First, a segmentation process is developed to classify a user's journey into GPS segments with only one mode of transport. Most researches have suggested modality inference models based on hand-built functionality, which can be vulnerable to traffic and environmental conditions. SECA combines a convolutional-deconvolutional autoencoder and a convolutional neural network into an architecture to perform supervised and unsupervised learning simultaneously.

In another paper (Dabiri et al., 2020), the same authors consider that transportation agencies are beginning to leverage the more available GPS trajectory data to support their analyses and decision-making. Although this representation of mobility data adds meaningful value to several analyses, a challenge is the lack of knowledge regarding the kinds of vehicles that produced the recorded tours, which restricts the value of the trajectory data in the transport system analysis. The paper presents a new design of GPS trajectories, which is compatible with deep learning models and also obtains vehicle movement features and

road features. To this end, an open-source navigation system is also applied to obtain more detailed information on travel time and the distance between GPS coordinates. The experimental phase shows that the proposed CNN-VC model consistently outperforms both classical ML algorithms and other essential DL methods.

The authors of (Zhang et al., 2019b) consider that, although some studies on the classification of trajectories have been conducted, they require manual selection of characteristics or fail to completely consider the influence of time and space on

the classification results. The features obtained are joined to provide the results of the final classification. Then, they present an approach based on the latest DenseNet image classification network structure and include the attention tool and residual





learning. This model can fully extract spatial features to increase feature propagation and capture long-term dependence. The results show that the design outperforms traditional models in terms of accuracy, recall, and f1 score.

The authors of the paper (Duan et al., 2018) consider the non-linear and space-time characteristics of urban traffic data, proposing a deep hybrid neural network enhanced by a greedy algorithm for the prediction of urban traffic flow using GPS tracking of taxis. The propose a deep neural network model that combines a convolutional neural network, which extracts spatial features, with long-term memory, which captures temporal information, to predict the flow of urban traffic. Experimental results based on real taxi GPS trajectory data from Xian City show that the enhanced deep hybrid CNN-LSTM model has higher classification accuracy and requires a shorter time than traditional methods.

Finally, based on GPS data the work presented in (Pierdicca et al., 2019b) shows that the case of urban parks is difficult, requiring the knowledge of many variables, which are difficult to consider simultaneously. One of these variables is the set of people who use the parks. This study aims to produce a method to identify how an urban green park is used by its visitors to provide planners and managing authorities with a standardised method. A trajectory classification algorithm is implemented to understand the most common visitor trajectories, obtaining the advantages of GPS and sensor-based traces. Based on these user-generated data, the proposed data-driven approach can determine the park's mission by processing visitor trajectories while using a mobile application specifically designed for this purpose.

## 3 Results and analysis

As mentioned previously, the use of AI in geomatics data management is not a new problem. Several studies have been conducted on this topic, and many are currently in development. Geomatics data are the core of several applications in which ML and DL have been applied.

The use of geographical and spatial information within society as well as in academic work has increased rapidly in recent decades. This also means that geomatics has started to create problems in both the academic and non-academic worlds. First, it bridges borders that have been in place for a long time, and second geomatics, or rather the basic concepts of geomatics, are increasingly being used. Spatial analysis has proven to be important in all disciplines. We can find examples of strong GIS units in, for example, humanities (archaeology, human ecology, language studies, etc.), social science (human and economic geography, economy, economic history, etc.), and medicine (social and occupational medicine, epidemiology, etc.). Thus, Geomatics is part of research in most disciplines, and many users are facing issues related to the integration of geomatics in their field. Geomatics is also used frequently in interdisciplinary settings, which leads to specific issues.

We close this paper by returning briefly to the questions raised at the beginning, which remain largely open.

**Comparing ML and DL, which is the most commonly used in geomatics?** First of all, it is necessary to make a clarification, as yet said known that DL is a type of ML approach, when following we compare DL and ML, we distinguish approaches that use DL from those that use ML except DL.

The comparison between ML and DL methods is shown in Figures 4 and 5. Figure 4 shows that the most used method, especially in the last years, is DL, with an average rate of 80% compared to a rate of 20% for ML.



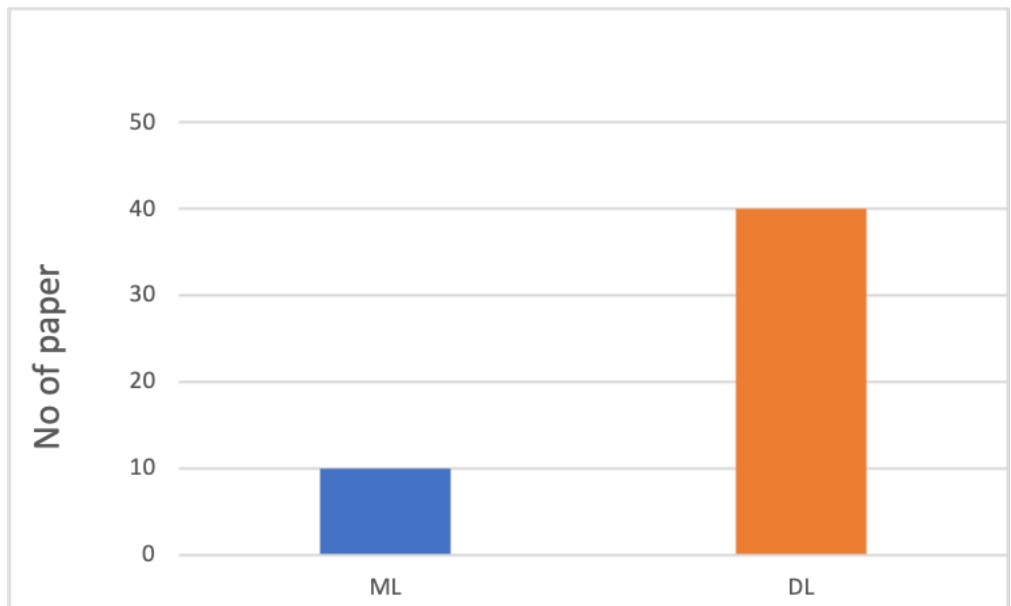

**Figure 4.** Comparison between ML vs DL approaches adopted among the reseach papers reviewed.

Figure 5 compares the two methodologies during the time interval considered, confirming that there is a greater use of DL than ML to deal with geomatics data in the period taken into consideration.

     Table 1 summarises ten of the applications reviewed for each kind of data, comparing the input, the task, and the AI method chosen.

Table 1: Brief review of information regarding the complex geomatics data analysed in this study, which are important for selecting an appropriate AI technique.

| Data | Method | Task | Application | Ref. |
|---|---|---|---|---|
| RGB-D 1 | DL | Semantic segmentation | Urban/rural scenes | (Li et al., 2017a) |
| RGB-D 2 | DL | Semantic segmentation | PV module | (Espinosa et al., 2020) |
| RGB-D 3 | ML | Object detection | Shadow detection | (Movia et al., 2016) |
| RGB-D 4 | ML | Classification | Rice plants | (Zheng et al., 2020) |
| RGB-D 5 | DL | Semantic segmentation | Building scenes | (Czerniawski and Leite, 2020) |
| RGB-D 6 | DL | Object detection | Urban/rural scenes | (Gong et al., 2018) |
| RGB-D 7 | DL | Object detection | Urban/rural scenes | (Duan et al., 2019) |
| RGB-D 8 | DL | Clustering | Urban/rural scenes | (Li et al., 2019b) |
| RGB-D 9 | DL | Semantic segmentation | Rice plants | (Yang et al., 2020) |
| RGB-D 10 | DL | Semantic segmentation | Urban/rural scenes | (Wang et al., 2017) |
| IRT 1 | ML | Object detection | Electrical equipments | (Ullah et al., 2017) |
| IRT 2 | DL | Object detection | PV module | (Akram et al., 2020) |
| | | | | Continue |





**Table 1 – continued from previous page**

| Data | Method | Task | Application | Ref. |
|---|---|---|---|---|
| IRT 3 | DL | Segmentation | PV module | (Luo et al., 2019) |
| IRT 4 | DL | Classification | Electrical equipments | (Nasiri et al., 2019) |
| IRT 5 | ML | Object detection | Electrical equipments | (Ullah et al., 2020) |
| IRT 6 | DL | Object detection | Electrical equipments | (Gong et al., 2018) |
| IRT 7 | ML | Classification | Electrical equipments | (Duan et al., 2019) |
| IRT 8 | DL | Object detection | PV module | (Li et al., 2018a) |
| IRT 9 | DL | Object detection | Composite material | (Al-Habaibeh et al., 2021) |
| IRT 10 | DL | Classification | Electrical equipments | (Chellamuthu and Sekaran, 2019) |
| PC 1 | DL | Classification | Building scenes | (Wang and Ji, 2021) |
| PC 2 | DL | Classification | Road marking | (Wen et al., 2019) |
| PC 3 | DL | Part segmentation | Building scenes | (Zhang et al., 2018) |
| PC 4 | DL | Classification | Complex forests | (Zou et al., 2017) |
| PC 5 | DL | Part segmentation | Indoor scenes | (Yousefhussien et al., 2018) |
| PC 6 | DL | Semantic segmentation | Urban/rural scenes | (Yang et al., 2017) |
| PC 7 | DL | Segmentation | Urban/rural scenes | (Wang et al., 2019) |
| PC 8 | DL | Semantic segmentation | Building scenes | (Pierdicca et al., 2020b) |
| PC 9 | DL | Semantic segmentation | Indoor scenes | (Qi et al., 2017a) |
| PC 10 | DL | Semantic segmentation | Urban/rural scenes | (Balado et al., 2020) |
| TRAJ 1 | DL | Classification | Urban/rural scenes | (Endo et al., 2016) |
| TRAJ 2 | ML | Object detection | Urban/rural scenes | (Habtemichael and Cetin, 2016) |
| TRAJ 3 | ML | Segmentation | Urban/rural scenes | (Jiang et al., 2017) |
| TRAJ 4 | ML | Classification | Urban/rural scenes | (Xiao et al., 2017) |
| TRAJ 5 | DL | Classification | Urban/rural scenes | (James, 2020) |
| TRAJ 6 | DL | Segmentation | Urban/rural scenes | (Dabiri et al., 2019) |
| TRAJ 7 | DL | Classification | Urban/rural scenes | (Dabiri et al., 2020) |
| TRAJ 8 | DL | Classification | Urban/rural scenes | (Zhang et al., 2019b) |
| TRAJ 9 | DL | Segmentation | Urban/rural scenes | (Duan et al., 2018) |
| TRAJ 10 | ML | Clustering | Urban/rural scenes | (Pierdicca et al., 2019b) |
| MSI/HSI 1 | ML | Classification | Urban/rural scenes | (Sharma et al., 2017) |
| MSI/HSI 2 | DL | Classification | Urban/rural scenes | (Zhong et al., 2017) |
| MSI/HSI 3 | DL | Classification | Urban/rural scenes | (Yang et al., 2018b) |
| MSI/HSI 4 | DL | Classification | Urban/rural scenes | (Wu and Prasad, 2017) |
| MSI/HSI 5 | DL | Classification | Urban/rural scenes | (Zhao and Du, 2016) |
| MSI/HSI 6 | DL | Object detection | Urban/rural scenes | (Li et al., 2017b) |
| MSI/HSI 7 | DL | Classification | Urban/rural scenes | (Liu et al., 2016) |
| MSI/HSI 8 | DL | Classification | Urban/rural scenes | (Chen et al., 2016) |
| MSI/HSI 9 | DL | Classification | Urban/rural scenes | (Hong et al., 2020b) |
| MSI/HSI 10 | DL | Classification | Urban/rural scenes | (Hong et al., 2020a) |



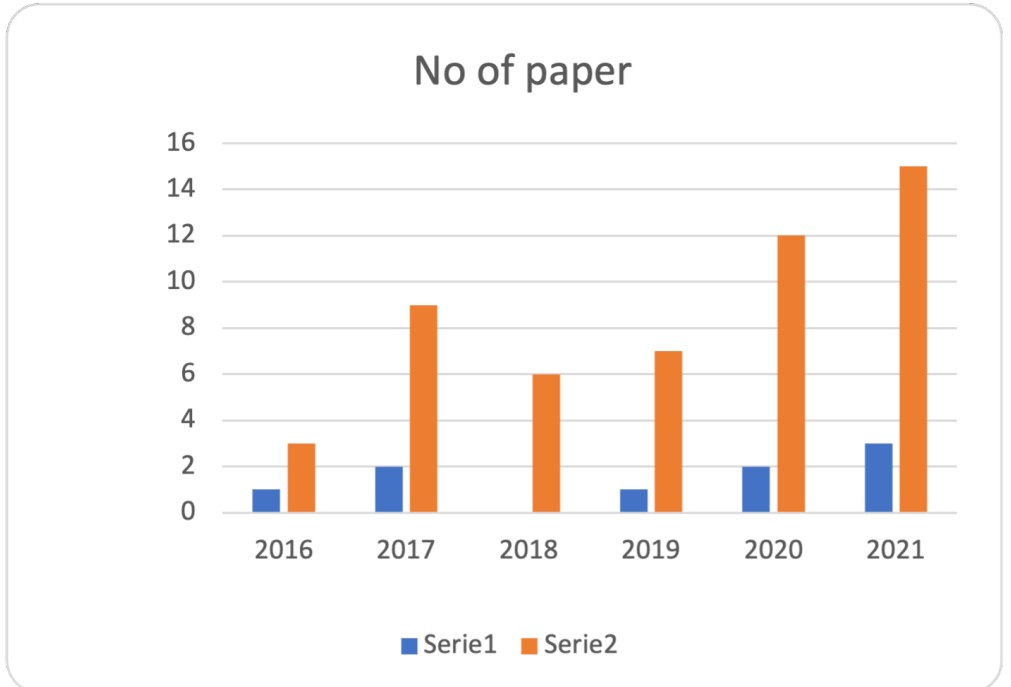

**Figure 5.** Comparison of ML vs DL approached, divided per year.

**Do geomatics data influence the choice of using one methodology rather than another?** Figure 6 shows the results in percentage terms. In the graphs, we have grouped the papers on geomatics data and the employed approach. For all data, the use of DL is gaining increasing importance, especially in point cloud semantic segmentation and classification. While for IRT data the use of DL techniques is slightly lower than with other types of data, this probably depends on the technical and physical characteristics of the IRT data. Thus, the use of one technology rather than another also depends on the type of data processed.

From these analysis it is also possible to answer the RQ1, as the data demonstrate the trend in preferring DL approaches rather than ML ones.

**For which tasks are geomatics data used?** The main tasks performed using geomatics data are shown in Figures 7 and 8. Observing Figure 7, the classification task is the most commonly employed, with a rate of 42%. The object detection task is employed 22% of the time, and the semantic segmentation task has a rate of 18%. The remaining 18% comprehends segmen-

tation, part segmentation and clustering. These results involve all geomatics data considered in this review. These data, which have different characteristics, mainly due to the type of acquisition, are used in tasks that can be included in the three identified categories.

Figure 8 considers the task referring to all types of data. Classification is the task mostly employed with HSI/MSI data, object detection seems to be the preferred solution when dealing with both IRT and RDB-D images. Meanwhile, the point

cloud data, confirming the trend from the literature review, are mainly used for semantic segmentation (with a rate of 40%),





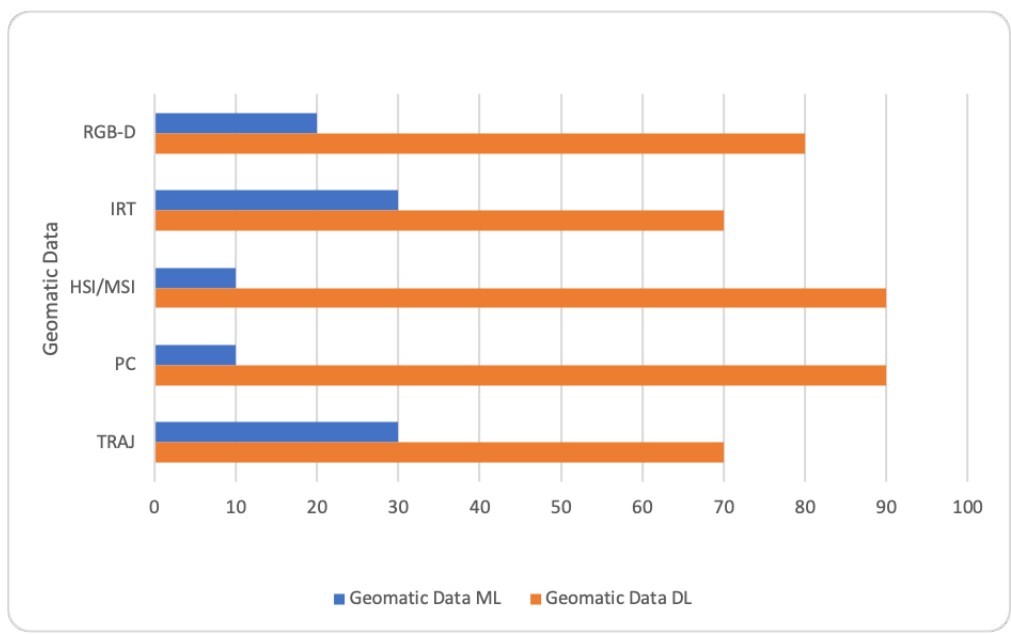

**Figure 6.** Comparison between ML vs DL approaches, selected according to the respective geomatics data for which they have been applied for.

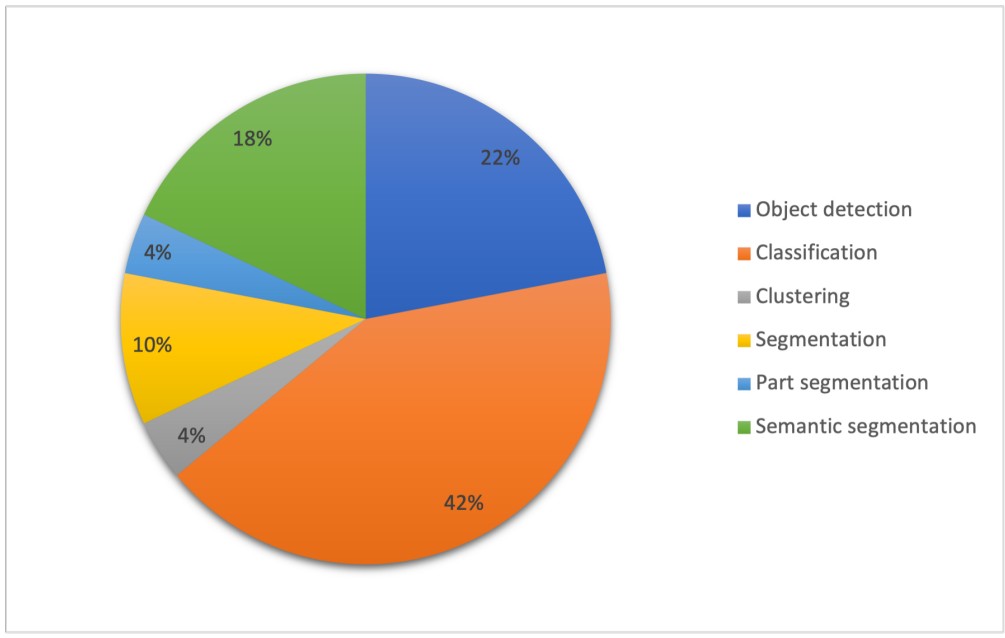

**Figure 7.** Distribution (percentage) of papers subdivided following the AI based task.





classification (with a rate of 30%) and part segmentation (with a rate of 20%) tasks. The object detection task is not executed. On the contrary the main task for IRT data is object detection with a rate of 60%, follows classification task with a rate of 30% and with a low rate segmentation (rate 10%). Classification and segmentation are the main tasks for the trajectories data. Other tasks are clustering and object detection with the same rate (10%).

This analysis has been fundamental to answer RQ2. Indeed, the AI approach is strictly connected with the kind of data, thus depending to the domain in which the approach is applied (see section 3).

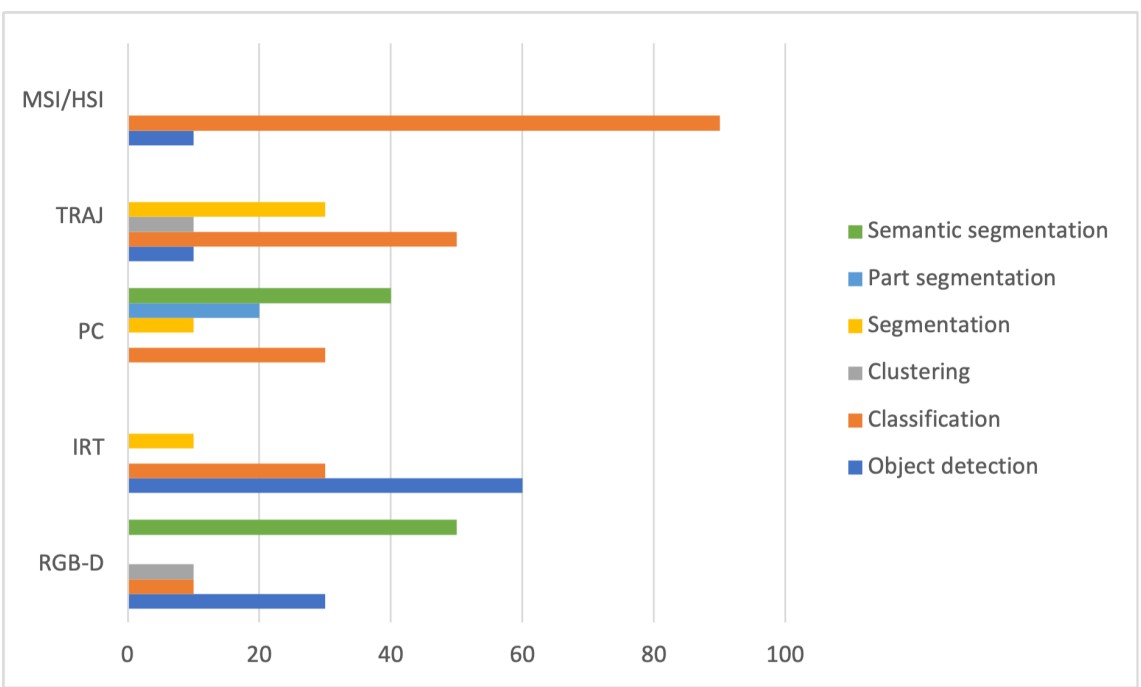

**Figure 8.** Comparison between Geomatics data and AI based tasks. The percentage of papers follow the criteria of matching, for each kind of data, the type of AI approach used.

**Are there relationships between application domains and geomatics data?** Figures 9 and 10 answer RQ3 and RQ4, which seeks to establish whether a relationship exists between the data type and application domains.

Taking into account the research mentioned in this paper, we have identified ten different aspects (urban/rural scenes, PV
module, shadow detection, rice plants, electrical equipments, composite material, road marking, building scenes, complex forests, and indoor scenes).

The analysis shown in Figure 9 considers each AI-based task based on the application domain. This graph directly comprises the application domain and geomatics data. RGB-D and PC data are most commonly used in different domains, although RGB-D data are most commonly used in urban/rural scenes. It is fair to say that a clear subdivision among the countless application
domains in Geomatic is impervious; notwithstanding, Figure 9 highlights that clustering and classification tasks are nowadays outperforming in Urban Scenes, maybe due to the vast use of geomatic data in such environments. PV plants applications





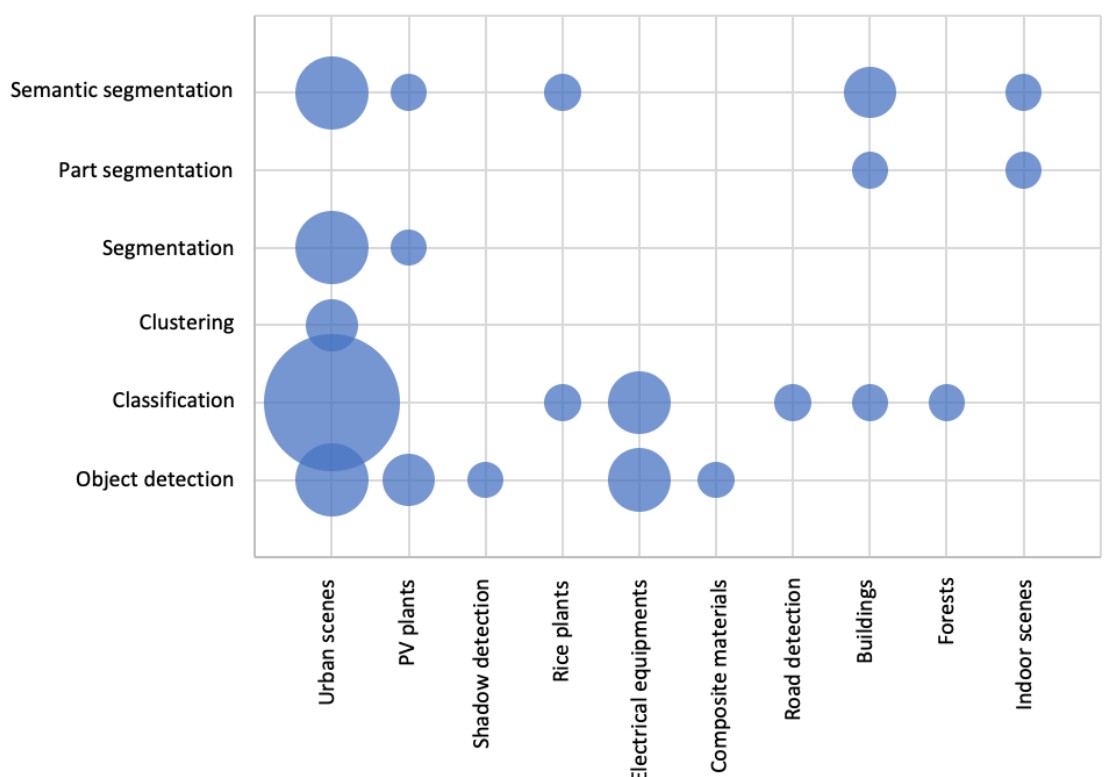

**Figure 9.** Relationship between specific application and AI based task.

are however explored, indicating that AI approaches might be very useful for decision making in environmental applications, as PV plants are. The remaining data are sparse, highlighting that there is the need for future investigations for outlining a straightforward line of research.

The analysis of Figure 10 raises an additional question: does the application domain change over the years? We can affirm that there is no relation between the application domain and the year, although there is an increase in the application of urban/rural scenes, mainly due to the type of associated data.

## 4    Discussion: Challenges, open issues, lesson learnt

Notwithstanding the success of AI in the geomatics, important caveats and limitations have hampered its wider adoption and 580    impact. Figure 11 presents a radar chart that considers the tasks based on the kind of data. This summarises the choice of task with available geomatic data.


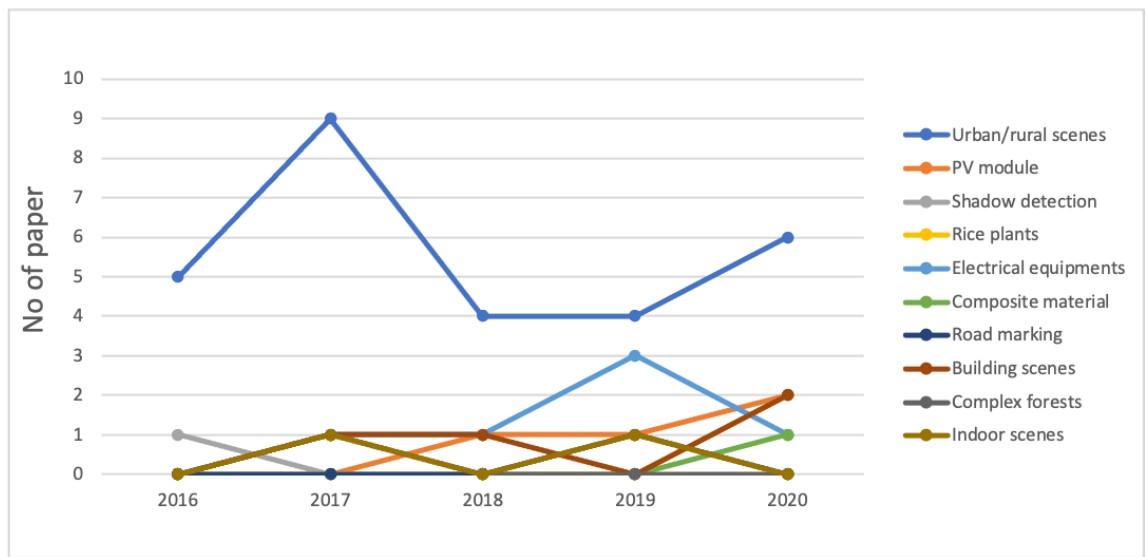

**Figure 10.** Number of papers for each application divided by year

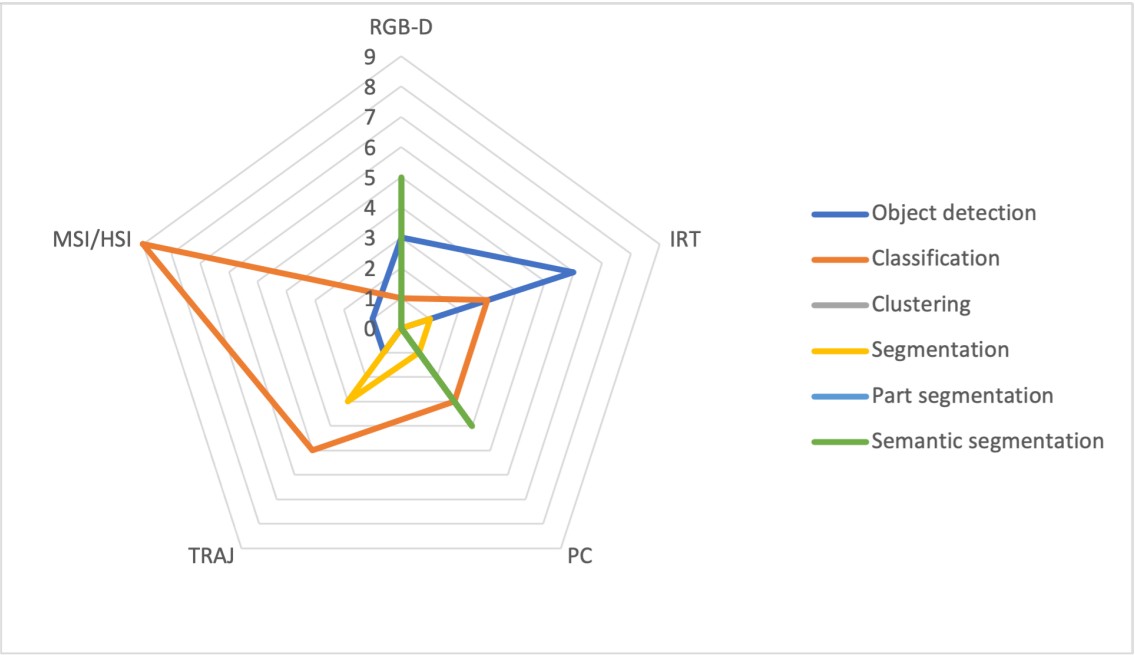

**Figure 11.** Spider chart representing the distribution of AI-Based approached in relation with the specific geomatic data.

Exploiting AI for the interpretation of complex geomatics data comes with many challenges, including the variability of the data source, the management of heterogeneous data, the different scales of representation, and the purpose of data processing. However, the more pronounced challenges related to application can be categorised as follows:





– **Lack of available dataset**: Regardless of the topic and/or the kind of data in the training phase (given the assumption that DL models can be arranged to fit a specific task), there is a lack of available datasets in the literature to be used as benchmarks. The great interest demonstrated by the research community in utilising geomatics data with learning-based approaches is hampered by the scepticism in sharing labelled datasets. It is well known that ML and DL are data-driven technique that perform better as the number of input samples increases. Attempts to solve this problem have involved

the generation of synthetic datasets (Pierdicca et al., 2019a; Morbidoni et al., 2020). Recently, generative models have proven to be effective for this task. Generative adversarial networks (GANs) are an appealing DL approach developed in 2014 by Goodfellow (Goodfellow et al., 2014). GANs are an unsupervised deep learning approach in which two neural networks challenge each other, and each of the two networks improves at its given task with each iteration. For the image generation issue, the generator begins with Gaussian noise to generate images, and the discriminator determines how

valuable the generated images are. This process proceeds until the generator development outputs. GANs have been used to generate artificial images and videos as well as to generate point clouds (Vondrick et al., 2016; Sun et al., 2020; Rossi et al., 2021). Despite exceptional results in supervised learning since the DL developments, collecting enough data to train the models remains a challenge, and some methods have been developed to train models with little or no data. Zero-Shot Learning (ZSL) is the task of training a model on some (seen) classes and testing it on other (unseen) classes.

Good results have been achieved in ZSL, especially with the adoption of generative methods, but it is unclear whether these results are generalizable to the real world. Moreover, self-supervision as an auxiliary task to the main supervised few-shot learning is considered to be a equivalent method to learn a transferable feature representation from limited examples, since self-supervision can contribute to additional structural information easily ignored by the main task.

        – **Domain dependant models**: Regarding its respective geomatics compartment, when there is no all-in-one solution for

every task, each AI-based model should be chosen according to the task one is attempting to solve. In other words, as AI improves, the need has emerged to understand how to make such models effective, choosing them according to the kind of data for which they have been designed. Integrating the knowledge of domain expert into AI models increases the reliability and the robustness of algorithms, making decisions more accurate. Moreover, the knowledge acquired for one task can be use to solve related ones thanks to transfer learning strategies. Transfer learning allows to leverage

knowledge (such as features, weights etc) from previously trained AI models for training newer models and even tackle problems like having less data for the newer task.

        – **Data preprocessing**: Broadly speaking, geomatics data have intrinsic features that make them very challenging for DL, especially convolutional neural networks. The reason for this is that AI is intended to utilise data that are ordered, regular, and on a structured grid. This means that data should be ordered, and pre-processing operations are still time consuming.

Actually, this represents one of the main bottlenecks, as it requires the presence of an expert for every single application domain.

        – **Hardware limitations**: despite the growing computational capabilities of better-performing CPUs and the advances in distributed and parallel high performance computing (HPC), the computational costs of the above-mentioned tasks





remain high. We are not still at a stage where the ratio between time/gained and resources/spent is in balance, making
the use of AI-based methods unhelpful at times compared with time-consuming but more affordable manual solutions.

## 5   Concluding Remarks

AI is thoroughly changing several application domains. In the geospatial domain, the data characteristics are particularly suitable for ML/DL approaches. Above all, ML/DL-based interpretation of 3D geomatics enables us to transcend explicit geospatial modelling and, therefore, to overcome complex, heuristics-based reconstructions and model-based abstractions. This paper
provides insight on new trends, techniques, and methods of GeoAI. In particular, a thorough survey of the literature related to the use of AI in geomatics and its methods has been presented, with a particular focus on ML and DL methods. Considering the last year, we can see that there was mainly an increase in RGB-D data and a small reduction of IRT data compared to the previous year. IRT data increased starting in 2017 until 2019, and then in 2020 it had a reduction. Trajectories and HSI/MSI data were mainly an object of research in 2016 and 2017, and then there was a reduction until 2020, when the topic received
renewed attention. There was an absence of IRT and PC in 2016, and this subject has been extensively studied, particularly in recent years. In fact, the advancing application areas of point cloud processing have already covered not only conventional fields in geospatial analysis, but also include civil engineering, manufacturing, transportation, construction, forestry, ecology, mechanical engineering and so on, becoming both more affordable, more versatile, thus more studied and examined. Specific emphasis has been given to RGB-D images, thermal images, HSI and MSI, and point clouds analysis and management. AI tech-
niques offer a promising solution to system development and rapid innovation. Further, AI approaches have addressed various challenges, such as point cloud classification, semantic segmentation object detection, and image classification. Methods and techniques for each kind of geomatics data have been analysed, the main paths have been summarised, and their contributions have been highlighted. The reviewed approaches have been categorised and compared from multiple perspectives, pointing out their advantages and disadvantages. Finally, several interesting examples of the GeoAI applications have been presented
along with input patterns, pattern classes, and the applied method. We are confident that this review offers rich information and improves the understanding of the research issues related to the use of AI with geomatics data, and moreover, helps to inform if and how AI methods and techniques could help the creation of applications in various fields. This paper thus paves the way for further research. Future research directions include the improvement of the algorithms to use other comprehensive features, thereby achieving better performance. Moreover, as these models were considered "black box" operators, they need to be inter-
pretable and explainable. In fact, the perception of DNNs as "black box" algorithms makes difficult to ethically justify their use in high-stake decisions, especially in case of failure. The adverse of black-box-ness is transparency, in fact it becomes difficult the search for a direct understanding of the mechanism by which a model works. Thus, the introduction of interpretability and explainability techniques are crucial, including visualization of the results for analysis by humans. Otherwise, domain experts would be hesitant to use techniques that are not straight interpretable, tractable and trustworthy, given the increasing request
for ethical AI.



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
