# Peer review of "GeoAI: a review of Artificial Intelligence approaches for the interpretation of complex Geomatics data"

_Geoscientific Instrumentation, Methods and Data Systems, 2021_

## Author Comment (AC1)

**Response to Referee #1**

**Response:** We thank the Reviewer for the comments and for the valuable suggestions. Our responses can be found in this response letter. We updated our manuscript adding new text in **red** (please refer to ***manuscript marked with changes.pdf***).

This paper presents an interesting review on the application of artificial intelligence AI algorithms (Machine learning ML and deep learning DL in particular) for processing and analyzing geomatics data. The authors considered in their review only the papers published between 2016 and 2021.

**Response:** *We appreciate your effort and attention in evaluating our paper and we thank the reviewer for his/her positive feedback.*

**R1.1:** Since the authors reviewed on ML and DL, I think that a brief introduction of these tools and especially the difference between them, will help the readers, that are not familiar to work with, to better understand why there is an increasing demand to use AI.

**Response:** *Thank you for this wise suggestion. We have added a brief description of ML, DL and their differences in paragraph 2.1 (lines 115-145).*

**R1.2:** I appreciated the way the authors took to describe the motivations of the work. But, I think that the fist question that we should ask is : Why researchers are increasingly interested to DL. Is it because of data complexity only? Efficacity or simplicity of these tools to implement?

**Response:** *Thanks to your comment we have the possibility to better detail the deduction that can be drawn by analyzing the research questions in the introduction. In particular, the main aspect that arises from our research is that the deep learning methods are increasingly often adopted for complex geomatics data analysis. This is due to the size of dataset available in the state of art and for the network architectures that automate feature learning without the need for manual extraction. The numerous layers in deep neural networks allow models to become more competent at learning complex features and performing more intensive computational tasks, i.e., accomplish many complex operations simultaneously. These aspects have been added together with relevant recent literature in the field (lines 145-160 and 165-170).*

**R1.3:** On what basis you have selected the "fundamental" sources of Geomatic data?

**Response:** This is a very interesting question by this reviewer. Thank you for that. Indeed, a categorization of geomatic data is a hard task and difficult to treat as a compartment. However, given the selected journals and considering their SJR, we noted that the categories of data described in Figure 1 are those mostly exploited for AI experiments. Moreover, considering the selection criteria of this review, we identified data representing physical models and phenomena that better fit with the AI tasks. Relevant literature has been added to stress this aspect (paragraph 2.2).

**R1.4:** In section 2.2.2, the authors cited the use of InfraRed Thermography IRT. First, please correct Thermography not termography.

**Response:** *According to the reviewer's suggestion, we corrected the typos.*

**R1.5:** You cited methods like Mask R-CNN, MLP or others. I was wondering why there is not the YOLO algorithm, it is one of the most used in object detection and segmentation in visual and infrared images.

**Response:** *We have cited the Mask region-based convolutional neural network (Mask R-CNN) since it can benefit from extra data, even if that data is unlabeled. Mask R-CNN is also capable for instance segmentation. We agree that Mask R-CNN takes more time for detection compared to YOLO that can be used in any kind of object detection in real time and can be considered as the better model between the mentioned two. However, these results are data specific and might change with changes in data distribution and Mask R-CNN architecture was adopted in several works because it simultaneously performs object detection and instance segmentation, making it useful for the automated inspection task. For this reason, several papers focusing on GeoAI adopt this network instead of YOLO. Notwithstanding, your comment is valuable and we added two important works which used YOLO for the detection accordingly:*

- *Greco, A., Pironti, C., Saggese, A., Vento, M., & Vigilante, V. (2020, January). A deep learning based approach for detecting panels in photovoltaic plants. In Proceedings of the 3rd International Conference on Applications of Intelligent Systems(pp. 1-7).*
- *Tajwar, T., Hossain, S. F., Mobin, O. H., Islam, M., Khan, F. R., & Rahman, M. M. (2021, May). Infrared Thermography Based Hotspot Detection Of Photovoltaic Module using YOLO. In 2021 IEEE 12th Energy Conversion Congress & Exposition-Asia (ECCE-Asia) (pp. 1542-1547). IEEE.*

**R1.6:** Besides, I would like to draw your attention that other researchers used image fusion to image preprocessing as a data enhancement method by fusing visible and infrared images. I raised these remarks since you have compared, in Fig 8, the percentage of papers that used geomatic data with AI and you have concluded in line 540 that IRT data is lower than other types of data.

**Response:** *Thanks for your suggestions and it could be useful to clarify this issue. The consideration on IRT data might appear misleading, but it refers only to the comparison with the type of data examined in this review. As stated in the introduction this work outlines AI-based techniques for analysing and interpreting complex geomatics data. In fact, Figure 1 summarizes and highlighted the purpose of this work, i.e. the definition of guidelines in which the reviewed approaches are categorised and compared from multiple perspectives, including methodologies, functions, and an analysis of the pros and cons of each category. Image fusion and multi task learning are increasingly adopted in several studies, however, these works are not useful for the guidelines definition but deserve investigations, thus we added this important aspect in our future works.*

**R1.7:** Please provide more accurate description of the improvements to the state-of-the-art knowledge.

**Response:** *Existing reviews explore particular approaches for analysing geomatics disciplines (e.g. remote sensing), generally based on Artificial Intelligence techniques to solve a specific issue. There are several examples of well-structured systematic reviews focused on this domain which are added in the introduction. The novelty of this work relies on the definition of guidelines in which the reviewed approaches are categorised and compared from multiple perspectives, including methodologies, functions, and an analysis of the pros and cons of each category. In fact, to the best of our knowledge, a complete review on GeoAI for deducing insights from geomatics data is not present in literature.*

**R1.8:** I have other general remarks:

- Please choose between American English or British English --> Analysing and analyzing for example

- The paper is not well revised. There are some grammatical and form errors, ex. line 175, 540… Also; please correct the legend of Fig. 5

**Response:** *We agree that the text needed a general revision. The paper underwent a professional proofreading, and the certificate is attached at the bottom of this letter. Now the text is consistent, and the minor glitches amended.*

[Figure]

**SCRIBENDI**

**Certificate of Editing and Proofreading**

This certifies that a version of the document titled

**Geomatics Review**

was edited and/or proofread by Scribendi as order number

**756074**

for clarity, consistency, and correctness according to the
requirements and guidelines specified by the client.

**Mon, 18 Jan 2021**

*Scribendi Inc*

**SCRIBENDI INC.**
405 RIVERVIEW DRIVE
CHATHAM, ON N7M 0N3 CANADA
+1 (519) 351 1626

www.scribendi.com

---

## Author Comment (AC2)

**Response to Referee #2**

**Response:** We thank the Reviewer for the comments and for the valuable suggestions. Our responses can be found in this response letter. We updated our manuscript adding new text in **red** (*manuscript marked with changes.pdf*).

**R.2.1:** The authors present a review of Artificial Intelligence (AI) approaches to propose a state of the art based on the analysis of which type of data, methodology and applications geomatics data are used.

**Global overview**

Firstly, the authors are thanked for their work which is well structured and well explained. The objectives of the paper are clear and the reading is eased thanks to a good paper organization. The authors have made an interesting analysis of the selected publications regarding many criteria that enlighten some trends. As a consequence, their analysis is deeply linked to their selection of papers which seems to represent a tremendous task. Even though such selection could be discussed and could lead to inconsistent trends, each topic is explained in detail.

However, the paper form needs to be reviewed.

1) Section 1.3 : Maybe the paper organization should come before Section 1.2? (particularly because Section 1.2 is cited in Section 1.3)

**Response:** *we thank the reviewer for the thoughtful comments and recommendations. We have carefully addressed the reviewer's suggestions and the introduction is substantially strengthened. We have addressed the reviewers' specific concern by moving Section 1.3 before Section 1.2.*

**R.2.2:** 2) Figures : make the figures homogeneous to help the reader. Sometimes there is a title inside the figure, sometimes not.

**Response:** *We have redesigned the figures accordingly.*

**R.2.3:** Moreover, make sure you have your axes labelled and that labels are set accordingly among the different figures (e.g. Figure 6 VS Figure 8: data types are not in the same order, y-axis label on Figure 6 and not on Figure 8), etc.

**Response:** *We have redesigned the figures accordingly.*

**R.2.4: Specific remarks / Questions**

1) An introduction of Machine Learning (ML) and Deep Learning (DL) and their differences would make sense in this paper, particularly because they are mentionned together many times.

**Response:** *We updated the manuscript according to the reviewer's comment. The updated manuscript includes new text and refined sentences in these directions.*
*In particular, a clear description has been added in section 2.1 ((lines 115-145)).*

**R.2.5:** 2) Could you explain how you selected the pertinent papers (l.87)?

**Response:** *Thanks for your suggestions and it could be useful to clarify in section 1.3 that the chosen guidelines follow the PRISMA workflow diagram. As stated in "Research strategy definition" we define guidelines for the review finalisation. These guidelines are motivated by the fact that artificial intelligence approaches for geomatics dataset are quite new. In particular, if we focus on generative adversarial neural networks (GANs) for GeoAI domain, the interesting paper starting in 2017. These lead to an exclusion of paper dated before 2016 for sake of completeness. Thus we claimed that: "The following sources of information were used in this study: ieeeXplore, Scopus, Sciencedirect, citepseerx, and SpringerLink. A set of keywords were chosen in relation to the Remote Sensing domain and based on preliminary screening of the research field. The keywords considered in the research initially were as follows: geomatics data, pattern recognition, artificial intelligence, machine learning, neural networks, supervised learning, unsupervised learning, statistical methods, Active learning, Imbalanced class learning, deep learning, Convolutional Neural Networks, classification, segmentation, detection, pattern recognition, applications, remote sensing data, hyperspectral data, point clouds data, RGB-D data, thermal data, and trajectory. To obtain more accurate results, the keywords were aggregated. In a set of queries, the keyword geomatics data was combined with others related to the methodologies (ML, DL, and more), and in other sets, remote sensing data were combined with the application (classification or detection). Each query produced a large quantity of articles, which were selected based on their pertinence and year of publication. Articles considered inconsistent with the research topic and published before the year 2016 were removed from the list. The temporal distribution of works dealing with geomatics data is shown in Figures 2 and 3. The papers considered for the review were published between the years 2016 and 2021. Figure 2 shows the temporal distribution of works dealing with AI for geomatics data. Figure 3 highlights the number of papers taken into consideration divided by the year of publication and by the type of geomatics data."*

**R.2.6:** 3) Also, have you been able to draw a quick history of the methods and data type/size used over the years that lead the community to this point?

**Response:** *This is a very interesting comment. Despite the difficulty, there are several reasons that led the community to this point. We have seen a very well motivated explanation in*
- *Reichstein, M., Camps-Valls, G., Stevens, B., Jung, M., Denzler, J., & Carvalhais, N. (2019). Deep learning and process understanding for data-driven Earth system science. Nature, 566(7743), 195-204.*
- *Mehonic, A., & Kenyon, A. J. (2022). Brain-inspired computing needs a master plan. Nature, 604(7905), 255-260.*

*These papers have been added to the text.*

**R.2.7:** This could answer the following question, inherent to your paper: Why researchers are using more and more DL?

**Response:** *Deep learning models are gaining much popularity due to their supremacy in terms of accuracy when training with huge amounts of data. In fact, with machine learning systems a human needs to identify and hand-code the applied features based on the data type (for example, pixel value, shape, orientation), a deep learning system tries to learn those features without additional human intervention. The main difference between the preference of applying deep*

*learning models instead of the machine learning once is that while standard machine learning models make insights without being explicitly programmed and improve their results progressively, they still need some guidance and adjustments from humans. Whereas, deep learning relies on neural networks. Deep Learning methods have been analyzed since given the huge amount of geomatics data Deep Learning methods achieve best performance both in terms of efficiency and time. These aspects have been added together with relevant recent literature in the field (lines 145-160 and 165-170).*

**R.2.8:** 4) In your research, how did you considered the papers that use the fusion of data and the combination of AI-based approaches?

**Response:** *Thanks for the advice because this point is crucial for our future works. Image fusion using deep learning framework has shown notable achievements in geomatics disciplines such as remote sensing. Moreover, we emphasize another aspect worth investigating, i.e. multi-task learning., which is a training paradigm in which machine/deep learning models are trained with data from multiple tasks simultaneously, using shared representations to learn the common ideas between a collection of related tasks. We aim to continue advancing the field now that we have understood its low-maturity but nevertheless promising nature and we highlight this important aspect in the discussions and conclusions .*

**R.2.9:** 5) In your conclusion, you make a comparison of the type of data used over the years (l.625-631). Is it based on Figure 3? If so, it means that this conclusion is dependant on the paper selection criteria.

**Response:** *Thank you for this remark. The discussions and conclusions have been drawn according to the research questions asked at the beginning of the paper; of course, all the diagrams (and not only figure 3) contributed to the identification of the major trends between geomatic data and AI-based tasks.*

**R.2.10:** Did you try to compare your result with the number of matches of your queries based on the keyword and year among the different sources of information?

**Response:** *A systematic review of the literature was conducted using PRISMA guidelines and electronic databases listed in our review. The sequel to a set of keywords was considered. They are chosen in relation to the geomatics domain and on the basis of a preliminary screening of the research field. To get more accurate results the keywords have been aggregated. In one set of queries, keywords deep/machine learning and geomatics were combined with methodology-related others, in other sets deep/machine learning and geomatics were combined with application. Each query produced a large amount of articles, which were selected based on relevance and year of publication. Articles found to be inconsistent with the research topic and published before the year 2016 were removed from the list.*

[Figure]

**SCRIBENDI**

**Certificate of Editing and Proofreading**

This certifies that a version of the document titled

**Geomatics Review**

was edited and/or proofread by Scribendi as order number

**756074**

for  clarity, consistency, and correctness according to the
requirements and guidelines specified by the client.

**Mon, 18 Jan 2021**

*Scribendi Inc*

SCRIBENDI INC.
405 RIVERVIEW DRIVE
CHATHAM, ON N7M 0N3 CANADA
+1 (519) 351 1626

www.scribendi.com